# Delivery of Nitric Oxide in the Cardiovascular System: Implications for Clinical Diagnosis and Therapy

**DOI:** 10.3390/ijms222212166

**Published:** 2021-11-10

**Authors:** Tianxiang Ma, Zhexi Zhang, Yu Chen, Haoran Su, Xiaoyan Deng, Xiao Liu, Yubo Fan

**Affiliations:** Key Laboratory of Biomechanics and Mechanobiology (Beihang University), Ministry of Education, Beijing Advanced Innovation Center for Biomedical Engineering, School of Biological Science and Medical Engineering, Beihang University, Beijing 100083, China; 17376355@buaa.edu.cn (T.M.); 18374353@buaa.edu.cn (Z.Z.); sy1910108@buaa.edu.cn (Y.C.); 13261677588@163.com (H.S.); dengxy1953@buaa.edu.cn (X.D.)

**Keywords:** nitric oxide delivery, computational modeling, flow-mediated dilation, NO release platform, inhaled NO therapy, stem cell therapy

## Abstract

Nitric oxide (NO) is a key molecule in cardiovascular homeostasis and its abnormal delivery is highly associated with the occurrence and development of cardiovascular disease (CVD). The assessment and manipulation of NO delivery is crucial to the diagnosis and therapy of CVD, such as endothelial dysfunction, atherosclerotic progression, pulmonary hypertension, and cardiovascular manifestations of coronavirus (COVID-19). However, due to the low concentration and fast reaction characteristics of NO in the cardiovascular system, clinical applications centered on NO delivery are challenging. In this tutorial review, we first summarized the methods to estimate the in vivo NO delivery process, based on computational modeling and flow-mediated dilation, to assess endothelial function and vulnerability of atherosclerotic plaque. Then, emerging bioimaging technologies that have the potential to experimentally measure arterial NO concentration were discussed, including Raman spectroscopy and electrochemical sensors. In addition to diagnostic methods, therapies aimed at controlling NO delivery to regulate CVD were reviewed, including the NO release platform to treat endothelial dysfunction and atherosclerosis and inhaled NO therapy to treat pulmonary hypertension and COVID-19. Two potential methods to improve the effectiveness of existing NO therapy were also discussed, including the combination of NO release platform and computational modeling, and stem cell therapy, which currently remains at the laboratory stage but has clinical potential for the treatment of CVD.

## 1. Introduction

Cardiovascular disease (CVD) is a major cause of human morbidity and death [1,2]. Nitric oxide (NO) has crucial roles in cardiovascular homeostasis, which occur in a dose-dependent manner. Both high and low local concentrations of NO may induce the development of CVD [3,4]. In the cardiovascular system, endothelial nitric oxide synthase (eNOS) is expressed in the endothelium (a single layer of cells that form the lining of blood vessels, of which dysfunction is thought to underpin most types of CVD [5,6,7]), and inducible nitric oxide synthase (iNOS) expressed at the inflammatory site, which mainly catalyze NO formation from L-arginine in the presence of molecular oxygen and reduced nicotinamide adenine dinucleotide phosphate (NADPH) [8]. Under physiological conditions, NO is mainly formed due to the catalysis of eNOS, depending on the mechanotransduction of wall shear stress (WSS) and endothelial function. Generated NO will quickly diffuse into adjacent vascular smooth muscle in the arterial wall and activate soluble guanylate cyclase (sGC), leading to the conversion of guanosine triphosphate (GTP) into cyclic guanosine monophosphate (cGMP). The subsequent activated cGMP-dependent protein kinases culminate in a decrease in intracellular Ca^2+^ concentration and the desensitization of the actin–myosin interaction in Ca^2+^, both of which lead to vasodilation [9,10] (Figure 1a). However, impaired endothelial function leads to the insufficient release of NO and an increase in the biochemical localization of atherosclerosis [11,12,13]. In arterial pathologies, such as atherosclerosis, anti-inflammatory M2-polarized macrophages in the lipid pool activate to the pro-inflammatory M1-polarized macrophages and express iNOS [14,15] (Figure 1b). Excessive NO, produced by iNOS, rapidly forms reactive nitrogen species (RNS), with the combination of reactive oxygen species (ROS) increasing the risk of plaque rupture [16]. Therefore, diagnostic indexes derived from the NO delivery process can potentially assess endothelial function and the development of atherosclerosis and other types of CVD.

In addition to endogenous delivery, recent advances in exogenous NO release and scavenging technologies have initially realized the modulation of the NO delivery process [17,18,19,20,21,22,23]. For example, the drugs as organic nitrates (e.g., nitroglycerin, isosorbide dinitrates, isosorbide mononitrate, etc.), inorganic nitrates/nitrites, and other donors can release NO [24] and L-NIL, L-NNA, and other scavengers can deplete NO [25]. In addition, inhaled NO therapy can induce vasodilation of the pulmonary vessels, aimed at improving oxygenation. Additionally, a few types of stem cells, such as mesenchymal stem cells (MSCs), are also capable of modulating the local NO delivery process via paracrine mechanisms [26]. However, due to the difficulties in accurately measuring and controlling the NO delivery process within the arteries, only a few products for the market of NO delivery—either diagnostic or therapeutic—have been approved by US Food and Drug Administration (FDA) so far [17,27]. In this tutorial review, we summarized the combination of flow-mediated dilation (FMD) [28,29], computational modeling [30,31,32,33], and novel NO bioimaging techniques that can assess NO delivery, thereby evaluating endothelial dysfunction and its derived CVD [34]. NO-related therapeutic methods, including the NO release platform, inhaled NO therapy, and potential stem cell therapy were also reviewed. We particularly focused on recent progress and the implications of assessing and modulating NO delivery in the above-mentioned clinical diagnostic and potential therapeutic methods.

## 2. The Assessment of Nitric Oxide Delivery as Diagnostic Tool

### 2.1. Diagnosing the Endothelial Function and Atherosclerosis Development through Computational Modeling of NO Delivery

The complex arterial mass delivery process is difficult to experimentally investigate in vivo, and computational modeling of the process has been widely accepted as a powerful tool for studying the delivery of low-density lipoproteins [35,36] and thrombus [37,38]. The unique advantage of computational modeling is the capability to control physiological parameters quantitatively; therefore, confounding factors that affect analysis can be eliminated. In this way, diagnostic indexes hidden behind clinical data can be extracted, and therapeutic strategies can be mechanistically guided [39]. However, due to the low concentration (nM–μM) and high chemical activity, the process of NO delivery is challenging to computationally model: the production, transmission (including diffusion and convection), and reaction processes are all nonnegligible and should be modeled simultaneously.

Pioneering works focused mainly on computationally modeling the diffusion and reaction patterns of NO delivery [40,41,42]. Then, Fadel et al. [43] and Plata et al. [44] modeled the processes of endothelial NO release and convection in a parallel plate flow chamber. In order to analyze NO delivery in arteries with disturbed flow, the delivery process, in a two-dimensional idealized stenosed artery, was computationally modeled. The model ignored the three-dimensional geometric complexity of an artery and considered the effects of a disturbed flow pattern caused by stenosis, the transmission, and reaction of NO in the lumen and arterial walls. This study found that NO concentration is significantly hindered distal to stenosis, which may lead to endothelial dysfunction [32]. The two-dimensional model was then extended to a three-dimensional physiological aorta, based on imaging data from magnetic resonance imaging (MRI). The results showed that low NO concentration is related to the occurrence of endothelial dysfunction and high-risk location of atherosclerosis, which is shown in Figure 2a [31]. The above computational models ignored the pathological effects of NO delivery. To assess the NO delivery process in atherosclerotic arteries, the recent model considers not only the arterial wall, but also the shape and the components of the atherosclerotic plaque, including NO production by activated macrophages in the lipid pool, inorganic calcification, and the breakdown of NO due to hemoglobin in the intraplaque hemorrhage. The result showed that NO is unevenly distributed at a high level around atherosclerotic plaque (Figure 2b), which potentially indicates the vulnerability of plaques [30]. In addition to the modeling of NO delivery in medium or large arteries, NO distribution in an ideal tubular arteriole was recently evaluated, which further considered the movements of red blood cells. The study illustrated that the permeability of red blood cells would largely determine the NO delivery process and atherosclerotic progression in microvessels [45,46].

Two computational approaches have recently been proposed to simplify the modeling of NO delivery. First, a reduced-order computational model was introduced to study the systematic delivery of NO in a coronary tree with multiple arterial branches, which simplified arteries through circuit connection of the resistor (R—represents the arterial resistance), inductor (L—represents the blood inertia), and capacitance (C—represents the compliance of the vessel). It was stated that abnormal NO delivery is associated with the damage of coronary arteries, which has potential to diagnose the occurrence of CVD [47]. Second, considering that endothelial NO production is highly associated with the mechanotransduction of wall shear stress (WSS), the group of Arzani analyzed WSS distribution in the lumen of arteries instead of directly calculating NO concentration [13,48]. They claimed that WSS magnitude dominates NO concentration at the luminal surface. High WSS magnitude is protective due to the clearances of atherogenic biochemical, and low WSS magnitude might promote endothelial dysfunction and atherosclerosis by localizing the atherogenic biochemical [49].

**Figure 2 ijms-22-12166-f002:**
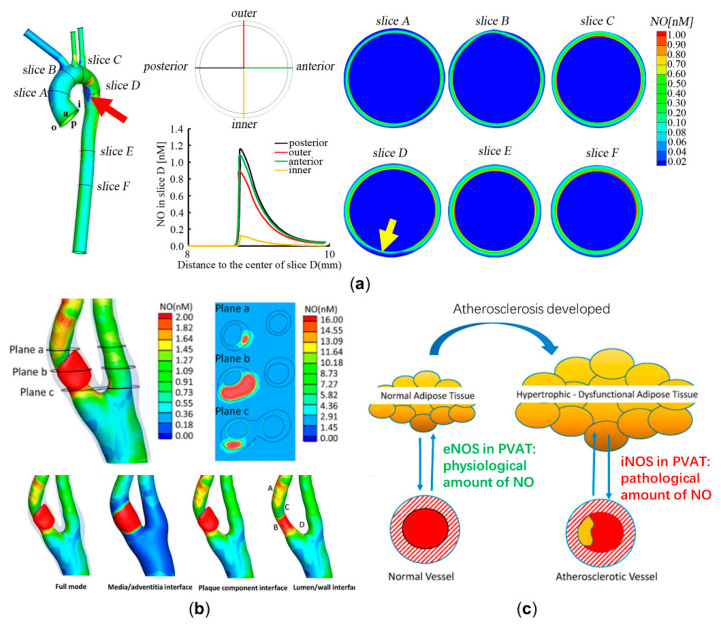
Computational modeling of nitric oxide (NO) delivery to assess atherosclerotic development. (**a**) NO distributions of six representative slices in the physiological aorta. NO distribution at slice D is the most uneven (a relative low NO concentration at the position indicated by a yellow arrow), corresponding to an atherosclerosis-prone site (the inner wall of the distal end of aortic arch has relative low NO concentration, which is marked by a red arrow) [31]. (**b**) A high NO concentration in the lipid pool is illustrated. NO distribution around atherosclerotic plaque is quite uneven, with the high levels at point A and point B and the low levels at point C and point D. (**c**) The effects of NO delivery by physiological and atherosclerotic perivascular adipose tissue (PVAT). Panel (**b**) is reproduced with permission from Reference [30], Elsevier. Panel (**c**) is adapted with permission from Reference [50], Elsevier.

In conclusion, based on computational modeling of NO delivery, the in vivo distribution of NO can be calculated to diagnostically predict endothelial dysfunction and atherosclerotic progression. However, aside from the lumen, arterial wall, and atherosclerotic plaque, more and more studies have shown that the perivascular adipose tissue (PVAT) could also affect NO delivery [51]. For physiological arteries, the eNOS expressed in PVAT would release moderate amounts of NO to stabilize endothelial function [52]. In arterial pathologies, the activated macrophages in inflamed PVAT would express iNOS and release massive amounts of NO, which may further increase the vulnerability of plaque by synergizing with the effects of the lipid core [50]. As far as we are concerned, investigating the exact role of PVAT in atherosclerosis using computational modeling may be a promising topic. A schematic diagram illustrating the role of PVAT in regulating endothelial function and atherosclerotic progression is shown in Figure 2c.

### 2.2. Improving the Specificity of the Flow-Mediated Dilation Test in Assessing Endothelial Function through Computational Modeling of Nitric Oxide Delivery

Flow-mediated dilation (FMD) refers to the physiological process that nitric oxide—synthesized and released by the endothelium activated by blood flow-induced shear stress—diffuses in the arterial wall and governs arterial dilation [28]. Through measuring flow- and shear-stimulated vasodilation, introduced by Celermajer, Deanfield, and colleagues [53,54,55], endothelial function can be assessed. Unlike the direct calculation of NO delivery in modeling, as reviewed in Section 2.1, FMD test aims to clinically measure vasodilation and reflect the overall NO delivery process. Technologically, the conduit artery, usually a brachial artery, is occluded using a sphygmomanometer cuff to supra-systolic pressures (typically between 200 and 300 mmHg) for 5 min distal for the ultrasound probe [29]. After the temporary ischemia ends when the cuff releases, the peripheral resistance reduces intensely and leads to an increase in blood flow and shear stress. The shear-induced NO, released by the endothelium, dilates the conduit artery, reflecting endothelial function. The magnitude of FMD test is conventionally described by the percentage flow-mediated dilation index (FMD%), calculated with the relative increase of examined artery, between the basal and maximal dilated moments. However, FMD% is not only determined by endothelial function, but is also affected by the large individual differences in measuring process, blood flow, blood pressure, and arterial diameter, limiting its diagnostic significance [28,29]. Two optimization strategies have been widely proposed to alleviate this problem. The first is to standardize the test procedure, which was recently proposed by Thijssen and colleagues [29]. The second is to normalize the measured FMD% to preclude interference factors other than endothelial function through clinical experiences and statistical methods, such as calculating the area under the curve of shear [56,57], analysis of covariance [58,59], and other methods [60,61].

Recent studies have drawn increasing attention to the exclusion of confounding factors other than endothelial function through computational modeling of the NO delivery process. It should be mentioned that vasodilation must be considered in addition to the above models, reviewed in Section 2.1, when modeling the FMD test. Yamazaki et al. pioneered the NO delivery computational model of FMD test [62]. The model has the ability to exclude the effect of individual differences in arterial stiffness from FMD% to enhance the specificity of endothelial function. The model considers molecular dynamics during NO delivery, including shear-induced NO release, changes in cyclic guanosine monophosphate (cGMP), Ca^2+^, and mechanosensitive channel activity [63]. Aside from modeling NO delivery from the small-scale mechanism, other studies modeled this process at a large scale. For example, the exposure–response model was established based on a simplified physical formula to describe NO delivery and vasodilation, which uniquely considers FMD time course and excludes the interference of shear exposure from endothelial function [64]. This model was recently extended to quantitatively reveal the interfaces of blood pressure and arterial stiffness on assessing endothelial function from FMD% [65,66].

In addition to the exposure–response model, Sidnawi et al. further modeled the vasodilating effect of NO by describing arterial stiffness as a function of WSS, and derived new parameters other than FMD% to represent the time response of arterial diameter and the arterial resistance to changing wall shear stress [67,68]. In addition, our group modeled the NO delivery process based on the advection–diffusion–reaction equation of NO transport, and considered deformation of the arterial wall, of which the stiffness is determined using NO concentration. The computational model has the ability to calculate personalized expected FMD% when the endothelium is completely healthy with the consideration of individual differences in blood flow, blood pressure, and arterial diameter. With the normalization of the measured FMD% through the expected FMD%, the differences between the real and healthy endothelial functions can be revealed, and the individual differences in the personalized data can be simultaneously excluded [69].

The above research mainly focused on modeling the NO delivery process from a mathematical point of view, and initially showed the potential to enhance the specificity of endothelial function in the FMD test. However, due to the bottleneck of NO measurement technology, current models inevitably rely on assumptions and lack direct verification, which limits clinical translation. For instance, endothelial NO release is assumed to be the same for each individual. Therefore, in Section 2.3, we focused on emerging bioimaging techniques that have the potential to experimentally measure NO concentrations in arteries.

### 2.3. Experimental Bioimaging of Nitric Oxide in Cardiovascular System

In Section 2.1 and Section 2.2, we reviewed methods to estimate the NO delivery process based on computational modeling and flow-mediated dilation test (Table 1). However, it is still challenging to experimentally measure NO concentration in arteries due to its low concentration and fast reaction characteristics. The widely used approach to is to measure the expressions of eNOS or iNOS through immunohistochemical methods to indirectly estimate NO distribution. For example, Qiao et al. experimentally measured sites of low eNOS expression in the aorta of rat, which is consistent with computational calculated sites of low NO concentration [70]. In addition, two experimental approaches have been proposed to directly measure NO concentration in arteries, including Raman spectroscopy and electrochemical methods.

Raman spectroscopy is a potential choice for experimentally measuring the NO delivery process by detecting the molecular structure without labeling. Surface-enhanced Raman spectroscopy (SERS) is a great choice to analyze molecules with a low concentration, shown in Figure 3a. Cui et al. designed a kind of reaction-based SERS nanoprobe for the detection of intracellular NO with *o*-phenylenediamine-modified gold nanoparticles. They utilized this technology to measure NO release by macrophages (iNOS) with a temporal resolution of 30 s and a sensitivity of 100 nM [71]. Following this work, Xu et al. designed a ratiometric SERS probe through the synthesis of compound 3,4-diaminobenzene-thiol, and achieved a NO sensitivity of 54 nM [72]. Recently, Chen et al. modified a SERS sensor with gold nanoparticles and synthesized 3,4-diaminophenylboronic acid pinacol ester to further enhance the NO detection range to the level of 0–105 nM. One of the features of their work is that peroxynitrite (ONOO^−^), synthesized by overproduced NO and reactive oxygen species (ROS), can be measured synchronously, which is the critical component increasing the rupture risk of plaque [73]. Since SERS is limited to in vitro measurement, a few studies have tried to use the fiber optic Raman spectroscopy to collect in vivo chemical signals (Figure 3b) [74,75]. However, due to the low level of concentration, NO measurement using fiber optic Raman spectroscopy has not yet been realized.

Electrochemical methods are more mature compared to Raman spectroscopy for NO delivery detection. Researchers are constantly working on modifying electrodes to better measure NO in various settings [76], the mechanism of which is transferring NO oxidation on electrode surface to a signal of measurable current. Pioneering work of monitoring the NO delivery process using different vasodilator drugs in the isolated guinea pig heart was completed by Fujita et al. [77]. The first achievement of in vivo measurement was a catheter NO sensor, which was used to measure the NO concentration in human coronary circulation, in real time, to evaluate endothelial function. The result showed that the plasma NO concentration in healthy subjects is significantly higher than that of patients with severe left ventricular dysfunction [78]. In addition, an acupuncture microsensor needle made of gold film, combined with iron porphyrin-functionalized graphene complex, was then proposed to monitor NO signal in rat via puncture [79]. The aforementioned electrochemical NO sensors are non-deformable and non-biodegradable, which may cause severe irritation and infection complications. Recently, Li et al. designed a flexible and degradable sensor to achieve real time measurement of NO delivery, with a wide detection range, high temporal resolution, and high biocompatibility, as shown in Figure 3c. They claimed that NO sensors could be further combined with electrical stimulators to realize multifunctional physiological assessments [80].

One of the main drawbacks of the electrochemical method is that it is limited to invasive assessment of NO delivery to the endothelium and cannot measure NO concentrations in arteries and atherosclerotic plaques. To fix this problem, computational modeling of NO delivery may be combined with experimental electrochemical measurements, which are capable of predicting the inside NO distribution, based on the measured value.

**Figure 3 ijms-22-12166-f003:**
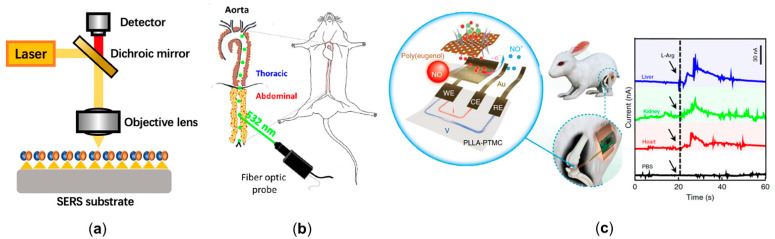
The emerging technologies to experimentally measure cardiovascular nitric oxide. (**a**) The schematic diagram of surface-enhanced Raman spectroscopy (SERS). A large enhancement in the Raman signal can be observed when nitric oxide (NO) is absorbed or lies close to the enhanced field at the surface. (**b**) The cardiovascular application of fiber optic Raman spectroscopy [75]. (**c**) The flexible and degradable electrochemical sensor to achieve real time measuring of NO delivery in rabbit, and the signal of NO concentration in various tissues [80].

**Table 1 ijms-22-12166-t001:** The available technological advancements in diagnostic measuring of NO delivery process.

Name of Authors	Applications of the Technology	Findings	Computational/Experimental
Liu et al. [31]	Assessing the early occurrence of endothelial dysfunction by NO distribution.	NO concentration at the inner wall of the distal end of aortic arch is significantly hindered, corresponding to the atherosclerotic prone site.	Computational
Qian et al. [30]	Assessing the development and vulnerability of formed atherosclerotic plaque by NO distribution.	The average NO concentration around the lipid plaque is significantly higher than the plaque-free region, which potentially indicates the vulnerability of plaque.	Computational
Arzani et al. [49]	Assessing the occurrence and development of atherosclerosis with wall shear stress, which can indicate NO delivery at endothelium.	Wall shear stress dominates the NO delivery process at endothelium, and the low wall shear stress indicates the occurrence and development of atherosclerosis.	Computational
Yamazaki et al. [62] Brackle et al. [64] Jin et al. [65,66]	Excluding the interferences of individual differences in arterial stiffness, shear exposure, and blood pressure from flow-mediated dilation (FMD) test.	The result of FMD is not only determined by endothelial function but is also influenced by the confounding factors. The computational modeling is effective to reduce these interferences.	Computational
Cui et al. [71]	Designing a reaction-based surface-enhanced Raman spectroscopy (SERS) nanoprobe for the detection of intracellular NO with *o*-phenylenediamine-modified gold nanoparticles.	The probe reaches a temporal resolution of 30 s and a sensitivity of 100 nM.	Experimental
Xu et al. [72]	Designing a ratiometric SERS probe with compound 3,4-diaminobenzene-thiol.	The probe enhances the NO sensitivity to 54 nM.	Experimental
Chen et al. [73]	Designing a SERS probe with gold nanoparticles and synthesized 3,4-diaminophenylboronic acid pinacol ester.	The probe further increases the NO detection range to 0–105 nM. And it is capable of detecting peroxynitrite (ONOO^−^) synchronously.	Experimental
Takarada et al. [78]	Using the catheter-type NO sensor to measure NO concentration in human coronary circulation.	Measured the NO delivery in coronary circulation for the first time and found that NO concentration in the patients with severe left ventricular dysfunction (2.3 nM) was significantly lower than normal subjects (12.0 nM).	Experimental
Tang et al. [79]	Designing an acupuncture microsensor needle by gold film and iron porphyrin-functionalized graphene complex.	The microsensor needle achieved the detection of NO signal in rat via puncture.	Experimental
Li et al. [80]	Designing a flexible and degradable sensor to realize real time measurement of NO delivery in vivo.	The sensor has a low detection limit (3.97 nmol), high temporal resolution (350 ms), and high biocompatibility.	Experimental

## 3. Nitric Oxide Delivery-Related Potential Therapeutic Approaches

Aside from assessing nitric oxide (NO) delivery to diagnose cardiovascular disease (CVD), the multiple functions of NO have also caused a great deal of interest in exploring solutions to therapeutically modulate NO release. The NO release platform is a direct approach to increase NO concentration in arteries. For arteries adjacent to the respiratory system (e.g., pulmonary vessels), inhaled NO therapy was proposed to strengthen the control of doses and locations of NO release. In addition, two potential methods are promising to further enhance effectiveness of NO therapy, including the combination of controllable NO release platform with computational modeling, and potential stem cell therapy.

### 3.1. Manipulating Nitric Oxide Delivery Process with NO Release Platform

The most widely used exogenous NO donors are organic nitrates (e.g., nitroglycerin, isosorbide dinitrates, and isosorbide mononitrate [24]); however, the features of unfavorable pharmacokinetics and the development of tolerance during chronic administration of organic nitrates might lead to increased synthesis of reactive oxygen species (ROS) and endothelial dysfunction [81,82,83]. To avoid the side effects of organic nitrates, inorganic nitrates or nitrites [84,85], nicorandil [86,87] and molsidomine [88] were gradually developed, showing promising effects to improve NO-mediated vasodilation and atherosclerotic plaque stability. Additionally, N-diazeniumdiolate (NONOate) has the unique advantage of having a high efficiency, which is capable of spontaneously releasing twice the amount of NO as a donor and has broad choices of half-life (2 s to 20 h) [17,89]. Incorporating polymers into a NO release platform allows delivering NO in a more continuous manner, including through micelles, dendrimers, star-shaped polymers, and polymeric nanoparticles [90]. Micelles use amphiphilic polymers to synthesize concentration-dependent structures through hydrophilic or hydrophobic interactions but have a low encapsulation efficiency [36,90,91,92]. Dendrimers are three-dimensional hyperbranched globular nanopolymeric architectures with the advantages of a narrow polydispersity index, controllable structure, and the availability of multiple functional groups at the periphery [93,94,95]. Star-shaped polymers show a more stable nature compared with micelles and are also easier to synthesize than dendrimers [90,96,97]. Polymer nano-, micro-, and milli-particles can encapsulate NO donors instead of covalent attachments, which can improve stability (Figure 4a) [98,99,100]. Synthesizing nanoparticles with NO donors and non-polymeric substances to deliver biomedical cargo has been attempted, including the metallic and nonmetallic nanoparticles [101,102,103]. For example, incorporating catalytic copper nanoparticles with S-nitroso-N-acetylpenicillamine can release NO from a blended donor and utilize endogenous donors in circulation (Figure 4b) [104], the modification of NO-release mesoporous silica nanoparticles with aminosilane can achieve appreciable levels of NO with tunable NO release durations [105], etc.

Because NO delivery must be administered in the correct dose, proper location, and time [83] to achieve an expected therapeutic effect, the clinical applications and efficacy of the abovementioned NO donors have been restricted due to difficulties in controlling drug release and achieving targeted delivery [17,27]. First, the in vivo release kinetics of NO donors vary greatly in in vitro experiments [106]. Second, the produced NO will diffuse randomly in the body, making it difficult to reach the expected spatial and temporal concentrations for the targeted tissue. Inversely, it may also cause adverse effects. For example, high concentrations of Fe_3_O_4_ nanoparticles that are surface-coated with aminoguanidine can increase NO levels to improve endothelium function, while low concentrations can induce a significant decrease in NO production and promote plaque vulnerability [107], showing potential toxicity to pathological arteries. Hence, the concentration of nanoparticles needs to be controlled within a safe range. In addition, taking into account individual differences, spatiotemporal distribution of NO in target tissues (such as atherosclerotic plaques) and in the expected healthy state varies greatly [30].

**Figure 4 ijms-22-12166-f004:**
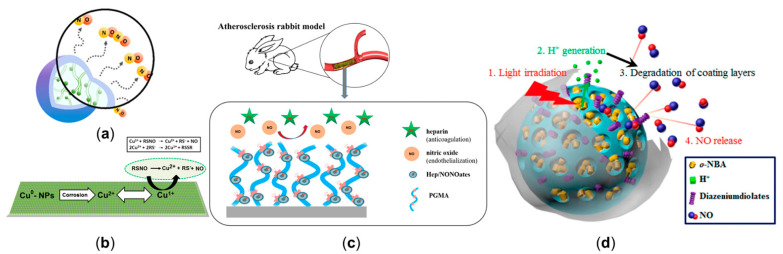
NO release platforms to manipulate cardiovascular nitric oxide delivery. (**a**) Branched polyethylenimine diazeniumdiolate (BPEI/NONOate) was encapsulated into PLGA nanoparticles to release NO in a sustained manner. (**b**) The mechanism of NO release from the combination of catalytic copper nanoparticles and S-nitroso-N-acetylpenicillamine. (**c**) The stent, coated with heparin/NONOate nanoparticles (Hep/NONOates), contributed to the long-term release of NO. (**d**) The light-responsive gatekeeper system for spatiotemporal-controlled NO delivery. Panel (**a**) is adapted with permission from Reference [108], American Chemical Society. Panel (**b**) is adapted with permission from Reference [104], American Chemical Society. Panel (**c**) is reproduced with permission from Reference [109], American Chemical Society. Panel (**d**) is reproduced with permission from Reference [110], American Chemical Society.

### 3.2. The Inhaled Nitric Oxide Therapy to Treat the Pulmonary Hypertension and Coronavirus (COVID-19)

Inhaled NO has the effects of relaxing pulmonary vessels and increasing the pulmonary blood oxygenation. It is the only NO-related method that has been formally approved by the US Food and Drug Administration (FDA), the European Medicine Evaluation Agency, and the Japanese Ministry of Health, Labour and Welfare (MHLW) for the clinical treatment of severe and persistent pulmonary hypertension of newborns [111]. The potential outcomes of inhaled NO therapy for adults have also been continuously investigated [112].

The main limitation of inhaled NO therapy is that the optimal dose threshold remains unclear, which can lead to long-term impairment, such as neurodevelopmental disabilities [113]. It is with this backdrop that Carter et al. designed an MRI-compatible NO delivery device, which realized simultaneous NO inhalation and measurement of vasodilation [114]. Schafer et al. further quantitatively investigated the correlation between inhaled NO and hemodynamic changes in intracardiac flow with four-dimensional-flow MRI [115]. They claimed that this method can link the quantity of delivered NO to the corresponding therapeutic effects, which may help the selection of the dose in clinical decision-making processes. The quantitative process of NO delivery and exchange in lung, based on the computational modeling, has also been explored [116]. However, systematic modeling of inhaled NO, NO delivery to trachea and arteries, and the vasodilating effects of NO have not yet been achieved.

Recently, the possible role of inhaled NO in coronavirus (COVID-19) therapy has received a great deal of attention [117]. In addition, FDA has recently granted expanded emergency use access for inhaled NO treatment of patients with mild or moderate COVID-19. Experimental and clinical evidence suggest that NO is potentially beneficial to the treatment of the coronavirus-mediated acute respiratory syndrome (e.g., acute cor pulmonale) due to its vasodilating effect. The results showed that more than half of subjects (*n* = 39) with COVID-19 that underwent inhaled NO therapy could avoid the use of invasive mechanical ventilators [118]. Tavazzi et al. also found positive results of inhaled NO therapy on dilating pulmonary circulation (*n* = 16) [119], which was supported by Lotz et al. (*n* = 7) [120]. In addition, the inhaled NO therapy may also prevent pulmonary vascular dysfunction, which is one of the major cardiovascular manifestations of COVID-19 [121]. However, the outcomes of this approach is still controversial; for example, Osama et al. indicated that inhaled NO therapy has little effect on treating acute cor pulmonale and might worsen the shunt related to pulmonary vasodilatation (*n* = 34) [122]. In addition, the proper dose of the inhaled NO is currently unclear. Hedenstierna et al. hypothesized that inhalation of NO in short bursts at high concentrations could be recommended to effectively prevent the progression of COVID-19 [123].

### 3.3. Potential Methods to Enhance the Effectiveness of Nitric Oxide Therapy

For the pulmonary vessels, doses and locations of NO released can be partially designed and controlled by inhaled NO methods as mentioned above (Section 3.2); however, for other vessels, it is still challenging to improve this difficulty. Two potential methods are promising to fix this problem, including the combination of a controllable NO release platform with computational modeling, and potential stem cell therapy.

#### 3.3.1. Combining Controllable NO Release Platform with Computational Modeling to Manipulate NO Delivery Process

In view of the shortcomings and limitations of the NO release platform mentioned above (Section 3.1), three pivotal steps might be needed to achieve the spatiotemporal delivery process of NO at the expected dose, location, and time, in line with the actual situations of a patient’s target tissues.

The first step is to analyze the NO delivery characteristics of the patient-specific target tissue and design the NO distribution that is expected to be achieved. As mentioned in Section 2.1, computational modeling can be utilized to assess the spatial distribution of NO in a target pathological region and the expected healthy state for each patient individually, so that the NO release platform can be targeted at regions with lower local NO concentrations compared with healthy states.

The second step is to use the platform to realize the designed NO transmission. Massive efforts have focused on developing controllable NO release platforms. For example, NO-release surface coatings can be applied to various supporting structures, such as stents (Figure 4c) [68,109,124] and metal–organic frameworks [125,126], to mimic endothelium function, thus promoting NO release and benefitting treatments of CVD. Such implants can maintain a high NO level at a specific site. Extensive attention is being put on stimuli-responsive systems, which are expected to fulfill on-demand NO delivery in a spatial-, temporal-, and dosage-controlled way [127]. Additionally, using a double trigger to control the release of NO (Figure 4d) [110] may be a feasible solution to guarantee the precision of the released location and dose. In addition, utilizing the prodrug strategy [128,129,130,131,132,133] of targeting NO release allows direct localized therapeutic delivery of a controlled dose via enzyme biocatalysis and provides a promising idea for the next generation of local targeted CVD therapy.

The third step is to analyze the NO release process to determine whether it is feasible. After the initial selection of the form of the platform, computational modeling can be used to simulate the kinetics of NO delivery, so as to obtain significant guidance for the optimized design of spatiotemporal delivery strategy to the expected target and improve the delivery efficiency. Taking nanoparticles as an example, the complex hemodynamic environment around a target tissue affects whether they can reach the target tissue at an expected dose and thus play their expected role [134,135]. By analyzing the blood flow topology, researchers have simulated a WSS-sensitive drug delivery system based on nanoparticle drugs, which may help to improve drug efficacy [136].

#### 3.3.2. The Clinical Potential of Stem Cell in Modulating Nitric Oxide Delivery

Stem cells refer to undifferentiated cells with indefinite division potential to differentiate into different types of cells and tissues [137], and are widely studied in regenerative medicine, including for restoring endothelial function and inhibiting inflammatory processes.

Mesenchymal stem cells (MSCs) are one of the most attractive therapeutic agents for various diseases [26,138,139]. Massive studies have illustrated that MSCs serve a positive role by improving local NO levels, thereby restoring endothelial function [26] (Figure 5a). Firstly, skin-derived MSCs can promote NO production by releasing vascular endothelial growth factor, thus enhancing the vasodilation [140,141]. Additionally, MSCs can enhance the eNOS level in oxidized low-density lipoprotein-exposed endothelial cells by secreting IL8/MIP-2 [142]. Secondly, for the developed atherosclerotic plaque with a lipid core, MSCs can not only improve eNOS expression of the endothelium, but also significantly decrease iNOS immunoreactivity by regulating macrophage polarization in inflammatory lipid cores, thereby reducing the risk of plaque rupture via paracrine mechanisms [14,143,144]. Additionally, MSCs can also differentiate into cardiovascular cells and promote angiogenesis [145]. Moreover, stem cell-derived exosomes (Figure 5a) also have the potential of protecting endothelial cells by promoting NO production [146,147,148,149,150,151,152]. Considering the roles of stem cells in regulating endothelial function by enhancing eNOS expression, and inhibiting inflammatory process by inhibiting iNOS expression, they might be regarded as an effective and powerful approach to treat CVD.

The clinical translations of NO-related stem cell approaches are hindered by various factors, such as the unknown dose range for the treatment of atherosclerosis [142]; furthermore, there is no consensus on the optimal selection of stem cells delivery methods [153]. Local delivery, a method of injecting cells into target tissues, is conducive to precise delivery, but it still has some limitations, such as the clinical risk of delivering massive amounts of cells to target tissues [154]. Combining stem cells with a NO release platform provides a novel regenerative medicine approach to improve a local NO delivery environment (Figure 5b) [27]. The NO released from the platform can interact with the stem cells and then improve the outcomes of stem cell therapy (Figure 5c) [155,156,157]. Through choosing the proper NO release platform, modulation of NO delivery by stem cells may be more effective.

**Figure 5 ijms-22-12166-f005:**
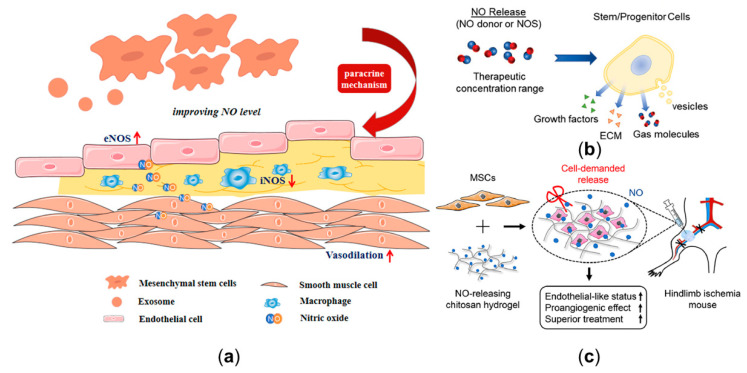
Nitric oxide (NO) delivery-related potential stem cell therapy. (**a**) The protective role of mesenchymal stem cells and exosomes in CVD by improving the eNOS expression of the endothelium and decreasing the iNOS expression of the macrophages. (**b**) The effect of NO release on stem cells. (**c**) The combination of mesenchymal stem cells with NO-release chitosan hydrogel-enhanced therapeutic function in hindlimb ischemia. Panel (**b**) is adapted with permission from Reference [27], Wiley Online Library. Panel (**c**) is adapted with permission from Reference [157], Elsevier.

## 4. Conclusions

Our knowledge of assessing the NO delivery process to diagnose cardiovascular disease is expanding. Despite recent advances in mathematical approaches to estimate the NO delivery process based on clinical images, the methods of directly measuring NO concentration in vivo still remain at preclinical stages. Aside from the assessment of NO delivery, the modulation of NO delivery has also aroused great interest, including the NO release platforms to directly release NO into the arteries, and inhaled NO therapy to treat the pulmonary hypertension and COVID-19. In addition, the use of stem cells to regulate NO delivery process may treat CVD by affecting the expression of iNOS and eNOS, but remains at the laboratory stage.

## Figures and Tables

**Figure 1 ijms-22-12166-f001:**
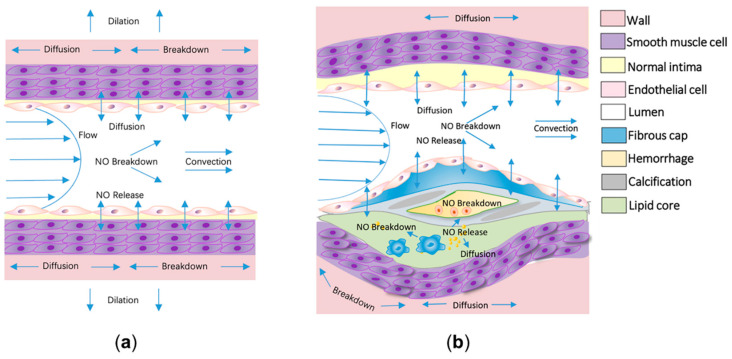
The delivery process of nitric oxide (NO) in physiological and atherosclerotic arteries. (**a**) At physiological sites, NO is mainly released by endothelial cells modulated by the shear flow. The released NO would be quickly delivered to neighboring arterial walls, leading to vasodilation. The diffusion, convection, and reaction processes of NO are shown in the schematic diagram. (**b**) In atherosclerotic arteries, in addition to the delivery process in physiological arteries, massive amounts of NO would release from the activated macrophages in the lipid core and the released NO would be synchronously consumed by hemoglobin in the intraplaque hemorrhage.

## Data Availability

Not applicable.

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
