# Peer review of "Delivery of Nitric Oxide in the Cardiovascular System: Implications for Clinical Diagnosis and Therapy"

_ijms, 2021, doi:10.3390/ijms222212166_

Round 1
Reviewer 1 Report
In this narrative review, Dr. Ma and colleagues discussed the available technologies for delivery of nitric oxide in the cardiovascular system. Overall, although it could be an interesting manuscript, there are some issues need to be addressed:
- The English needs to be significantly improved because it impairs the readability and clarity of this manuscript. Please consult with a native English writer or professional editing service.
- "The eNOS expressed in endothelial cells is the main source of NO in physiological arteries" How can an enzyme be a source of a substance? I am confused.
- "NO would fast diffuse into the adjacent arterial wall, and lead to the vasodilation by relaxing vascular smooth muscle" Please explain how NO can lead to SMC relaxation.
- "Besides, drugs as organic nitrates, inorganic nitrates/nitrites and other donors can release NO[25], L-NIL, L-NNA and other scavengers can deplete NO[26]" I don't see isosorbide dinitrates (ISDN) or mononitrate (ISMN) or nitroglycerin in the discussion. Why? They are very common in acute CV care as one of the first drugs given to ACS patients. Please add the discussion about them.
- "However, due to the difficulties in the accurate measurement and regulation of arterial NO delivery process, few US Food and Drug Administration (FDA)-approved diagnostic and therapeutic methods have been proposed for CVD up to now" What does this sentence mean? maybe for detecting NO in CVD or something?
- "The assessment of nitric oxide delivery as therapeutic tool" What does this mean? Which one is the therapeutic tool?
- Please explain about the definition of "idealized artery"? What is that?
- Lines 82-101: I would suggest to explain everything in a constructive way. What was the objective? How was it done? What are the findings and What are the conclusion of these studies. I tried to digest this paragraph multiple times but I still don't get the idea.
- Figure 2A: The uneven distribution of NO in slice D needs to be marked by star or arrow or box. Anything to show the readers what and where to see in those slices.
- Figure 2A: in the 3D image of the aorta, please mark the location of the atheroscerosis as well.
- Adding a table summarizing the available technological advancement to date would be very useful. The authors can include such information: Name of authors, applications of the technology, findings, computational/experimental. Please add it.
- Regarding the discussion about inhaled NO in COVID-19. Please also add the number of samples used in each studies and the study design, also a brief result because they might not be true.
- "The implication of nitric oxide delivery manipulating with stem cell related treatment" What does this mean? Please rephrase
- Also, I am not sure if stem cell is currently a proven therapy for CVD. So, I don't know why it is inside the therapeutic approach section. Please clarify.
- I think section 3 in general needs to be restructured. Otherwise please justify the argument of grouping them together.
Once again, I would strongly advise the authors to seek a help from a native English scientific writer to rewrite most parts of the manuscript. Once it is rewritten, maybe it can be more understandable on what the authors tried to deliver specifically in this manuscript.
Reviewer 2 Report
The delivery of nitric oxide in the cardiovascular system: implication for clinical diagnosis and therapy.
Dear Editor, thank you for the opportunity to review this article.
This is a review article with the main subject being the use of NO in medicine.
The only note I make is that it is too self-referential.
Round 2
Reviewer 1 Report
Thanks for addressing my comments and suggestions. I do still have some suggestions to improve the quality of this manuscript:
- I think somewhere in the introduction (around line 100), the authors could explain briefly the benefit of computational modeling in cardiovascular research, particularly to improve the mechanistic understanding of physiology and to guide therapy. It is also important to note that computational modeling has 100% control over parameters, which could be useful to do those things. The authors could check this translational publication about the role of modeling (PMID: 33890620).
- "The implication of nitric oxide delivery analysis in diagnosing the endothelial function and atherosclerosis development", I don't think this is understandable. The implication of analysis? Please rephrase.
- "The implication of nitric oxide delivery assessment in flow-mediated dilation", similarly, this heading needs to be rephrased. I am confused because an assessment cannot influence biological system? How could it impact flow-mediated dilation?
- Overall, I would suggest to check everything all over again because I feel that this manuscript is still not easy to read. The storyline needs to be refined to improve readability. I feel that the titles of the sections could be clearer. Sometimes, I still misinterpret the computational data with experimental data because there is no clear cut. Please check again.
